# Polymorphisms and gene expression in the almond IGT family are not correlated to variability in growth habit in major commercial almond cultivars

**Álvaro Montesinos** [1,2], **Chris Dardick**[3], **María José Rubio-Cabetas**[1,2], **Jérôme Grimplet** [1,2]*

**1** Centro de Investigación y Tecnología Agroalimentaria de Aragón (CITA), Unidad de Hortofruticultura, Gobierno de Aragón, Avda. Montañana, Zaragoza, Spain, **2** Instituto Agroalimentario de Aragón–IA2 (CITA-Universidad de Zaragoza), Calle Miguel Servet, Zaragoza, Spain, **3** Appalachian Fruit Research Station, United States Department of Agriculture—Agriculture Research Service, Kearneysville, WV, United States of America

\* jgrimplet@cita-aragon.es

**Data Availability Statement:** All relevant data are within the manuscript and its Supporting Information files.

## Abstract

Almond breeding programs aimed at selecting cultivars adapted to intensive orchards have recently focused on the optimization of tree architecture. This multifactorial trait is defined by numerous components controlled by processes such as hormonal responses, gravitropism and light perception. Gravitropism sensing is crucial to control the branch angle and therefore, the tree habit. A gene family, denominated IGT family after a shared conserved domain, has been described as involved in the regulation of branch angle in several species, including rice and Arabidopsis, and even in fruit trees like peach. Here we identified six members of this family in almond: *LAZY1*, *LAZY2*, *TAC1*, *DRO1*, *DRO2*, *IGT-like*. After analyzing their protein sequences in forty-one almond cultivars and wild species, little variability was found, pointing a high degree of conservation in this family. To our knowledge, this is the first effort to analyze the diversity of IGT family proteins in members of the same tree species. Gene expression was analyzed in fourteen cultivars of agronomical interest comprising diverse tree habit phenotypes. Only *LAZY1*, *LAZY2* and *TAC1* were expressed in almond shoot tips during the growing season. No relation could be established between the expression profile of these genes and the variability observed in the tree habit. However, some insight has been gained in how *LAZY1* and *LAZY2* are regulated, identifying the *IPA1* almond homologues and other transcription factors involved in hormonal responses as regulators of their expression. Besides, we have found various polymorphisms that could not be discarded as involved in a potential polygenic origin of regulation of architectural phenotypes. Therefore, we have established that neither the expression nor the genetic polymorphism of IGT family genes are correlated to diversity of tree habit in currently commercialized almond cultivars, with other gene families contributing to the variability of these traits.

**Funding:** Research was funded by the grants RTA2014-00062-00-00 (MJRC, JG, AM), RTI-2018-094210-R-100 (MJRC, JG, AM) and FPI-INIA CPD2016-0056 (AM) from the Spanish Research State Agency (AEI) (https://www.ciencia.gob.es/portal/site/MICINN/aei). The funders had no role in study design, data collection and analysis, decision to publish, or preparation of the manuscript.

**Competing interests:** The authors have declared that no competing interests exist.

## Introduction

In the last decade, more intensive almond orchards have become the predominant model in the Mediterranean areas, in order to increase productivity and to reduce labor cost [1]. Under this scenario, there is a growing interest in developing almond cultivars more adapted to mechanical pruning and presenting a natural branching that reduces pruning cost to achieve the desired tree structure. In consequence, optimized cultivars need to have low vigor, reasonable branching and an upright overall architecture.

Tree architecture is a highly complex trait defined by the sum of phenotypic components that influence the three-dimensional shape of the tree. It involves growth direction, growth rhythm, branching mode, position of the branches, the sexual differentiation of meristems and the length of axillary shoots [2]. Tree architecture is affected by environmental parameters such as light perception, gravity sensing, sugar availability or nutrients supply that take part in the plant physiological and hormonal regulation [3–5].

Two physiological processes that affect the plant architecture are apical dominance and the lateral bud outgrowth. Auxins act as the principal factor in the control of apical dominance. This hormone is synthesized at the apical leaves and transported throughout the plant, inhibiting lateral bud outgrowth. It promotes strigolactone (SL) biosynthesis, which is able to translocate to the bud and stop bud outgrowth [6, 7]. Cytokinins (CKs) act antagonistically to SLs, promoting Shoot Apical Meristem (SAM) differentiation and therefore bud outgrowth [8, 9]. Sugar availability has also been described as a positive regulator of bud outgrowth [10, 11]. These processes are essential for shaping the plant structure, although the overall tree habit, which is defined by the relative angle of the branches, is essentially regulated by two responses: light perception and gravitropism.

Light perception regulates both the growth and the direction of lateral branches. It is based on the ratio between red light and far red light (R:FR), captured by phytochrome photoreceptors phyA and phyB. When the R:FR is low, phyA is activated while phyB is inhibited, which sets off the inhibition of bud outgrowth, redistributing the auxin flux and focusing plant efforts in the growth of the primary axis [12–15].

Gravitropism is the main regulator of the branching angle. Its regulation occurs in specific cells called statocytes, where organelles containing large starch grains, called amyloplasts, act as gravity sensors [16]. These organelles sediment in the direction of the gravitational vector, triggering a signal which involves the opening of ion channels and the reorganization of the cytoskeleton [17–19]. This response leads to a relocation of auxin carriers PIN3 and PIN7 changing the direction of the auxin flux, which provokes a differential growth and a curvature in the opposing direction of the gravitational vector [20–22].

*LAZY1* has been described extensively as an influential factor in the control of plant architecture since its characterization in *Oryza sativa* (rice) as a regulator of tiller angle in agravitropic mutants [23–25]. Orthologs of this gene were found in *Arabidopsis thaliana* and *Zea mays* (maize), leading to the characterization of the same family in these species [26–28]. This family also includes *DRO1*, which was initially reported as an influential factor of root architecture in rice [29, 30]. *LAZY1* is related to *TAC1*, which is also involved in plant architecture regulation. *TAC1* was first identified in rice mutants with increased tiller angle, and it has also been characterized in Arabidopsis [31, 32]. *TAC1* differs from the rest of the family, denominated IGT family, in its lack of an EAR-like conserved domain denominated CCL domain located in the C-terminal region, which consists of 14 aminoacids [31, 33]. This conserved region is essential for the function and subcellular localization of IGT proteins. Since *LAZY1* and *TAC1* promote opposite phenotypes, and due to the lack of the CCL conserved domain, *TAC1* has been proposed as a negative regulator of *LAZY1* activity, in an upstream capacity [31, 33, 34]. However, the specific mechanism of the interaction between *LAZY1* and *TAC1* interaction is yet to be discovered [35].

The involvement of IGT family genes in gravitropism has been described in Arabidopsis and rice, acting as mediators between the sedimentation of statoliths gravity sensors and the relocation of auxin PIN carriers [33, 36–38]. Although a direct interaction with the phyA-phyB system is yet to be discovered, *TAC1* expression is influenced by the light perception regulator *COP1*, which would provide for integration between light and gravity responses [39].

The analysis of the mutation *br* in *Prunus persica* (peach), which is related to vertically oriented growth of branches, led to the annotation of an ortholog of *TAC1* [31]. Further studies have described the involvement of *TAC1* in auxin response mechanisms within different branching genotypes in peach, proving that the mechanisms involved in the control of the growth habit are conserved to a certain point in *Prunus* species [40, 41].

A total of 6 members of the IGT family have been found in *Prunus dulcis*: *LAZY1*, *LAZY2*, *DRO1*, *DRO2*, *IGT-like*, *TAC1*. With the exception of *TAC1*, all of them have the five conserved regions described in Arabidopsis [33]. In this study we carried out a genomic comparison for these six genes in forty-one almond cultivars and wild species with different growth habit phenotypes. Moreover, we analyzed the gene expression of the IGT family members in fourteen selected cultivars and searched for variants in their promoter region. Posteriorly, *LAZY1* and *LAZY2* promoters were inspected to identify regulatory elements (REs) associated to transcription factors (TFs) that could be involved in the regulation of *LAZY1* and *LAZY2*. Twenty-one TFs were selected due to its described function or its presence in growing shoot tips in previous studies and the analysis of their gene expression was carried out.

## Material and methods

### Almond tree populations

Forty-one cultivars and wild species (S1 File), whose genome had been previously obtained as part of the almond sequencing consortium [42] were selected to perform the comparative analysis of the IGT family protein sequences (S2 File). From these, twenty-seven cultivars were phenotyped for growth habit (S1 File), using a scale from 1 to 5 according UPOV guidelines: 1 = upright ($< 60˚$), 2 = somewhat upright ($60˚$ - $80˚$), 3 = semi-open ($80˚$ - $100˚$), 4 = open ($100˚$ - $120˚$), 5 = weeping ($> 120˚$) [43]. Fourteen cultivars of agronomical interest were selected to analyze the gene expression of the IGT family members. Ten out of these fourteen were chosen to analyze the expression of twenty-one transcription factors (Table 1).

### Comparative genomics

The cultivar genomes were assembled against the *P. dulcis* Texas Genome v2.0 [42] (https://www.rosaceae.org/analysis/295). Adapter sequences were removed by processing the raw reads sequences of the 41 cultivars with Trimmomatic v0.36.6 [44]. Alignments were performed using the Bowtie2 package (Galaxy Version 2.3.4.3) [45, 46]. Variant calling to detect SNPs was performed with the FreeBayes package (Galaxy Version 1.1.0.46–0) [47]. SNPs were filtered with the PLINK package (Galaxy Version 2.0.0) [48, 49] using the following parameters: read depth (DP) = 10; alternated allele observation count (AO) = 0.2. Promoter regions of the IGT family members were analyzed up to 2,000 pb upstream the start codon. All procedures were carried out using the Galaxy platform.

### Phylogenetic tree

The evolutionary history was inferred by using the Maximum Likelihood method and Poisson correction model [50]. The tree with the highest log likelihood (-5447.29) is shown. Initial tree (s) for the heuristic search were obtained automatically by applying Neighbor-Join and BioNJ

**Table 1. List of cultivars selected for the gene expression analysis of the IGT family members.**

| Cultivar | Tree habit |
|---|---|
| 'Forastero' (FOR) | Upright |
| **'Bartre' (BAR)** | Upright |
| **'Ferragnes' (FER)** | Somewhat upright |
| **'Garfi' (GAR)** | Somewhat upright |
| **'Garnem' (GN)** | Somewhat upright |
| **'Diamar' (DIA)** | Somewhat upright |
| **'Marinada' (MAN)** | Somewhat upright |
| 'Soleta' (SOL) | Semi-open |
| 'Marcona' (MAC) | Semi-open |
| **'Vairo' (VAI)** | Semi-open |
| **'Isabelona' (ISA)** | Semi-open |
| **'Vialfas' (VIA)** | Semi-open |
| 'Guara' (GUA) | Open |
| **'Desmayo Largueta' (DLA)** | Weeping |

The ten cultivars in bold were posteriorly chosen to study the expression of transcriptions factors associated to *LAZY1* and *LAZY2* promoters. Overall tree habit phenotype for each cultivar is described categorically according UPOV guidelines.

algorithms to a matrix of pairwise distances estimated using a JTT model, and then selecting the topology with superior log likelihood value. This analysis involved 252 amino acid sequences. There were a total of 424 positions in the final dataset. Evolutionary analyses were conducted in MEGA X [51].

## Quantitative real-time PCR (qPCR)

Tissue samples for the fourteen selected cultivars were gathered at the same day from adult trees at the end of summer (late August), when one-year old branches were developed, while maintaining an active growth. Cultivars were kept at an experimental orchard in Centro de Investigación y Tecnología Agroalimentaria de Aragón (CITA) (41˚43'29.4" N 0˚48'27.3" W). Five cm of the tip from one-year old lateral branches were collected. Each biological replicate consisted of three tips from the same tree. RNA extraction was performed from these samples using the CTAB method described previously [52] with some modifications [53–55]. Extracted RNA was quantified using a NanoDrop® ND-1000 UV-vis spectrophotometer (NanoDrop Technologies, Wilmington, DE, USA). RNA integrity was verified by electrophoresis on a 1% agarose gel. RNA samples (2500 ng) were reverse transcribed with SuperScript III First-Strand Synthesis System (Thermo Fisher Scientific, https://www.thermofisher.com) in a total volume of 21 μL according to the manufacturer's instructions. qPCR was performed using the Superscript III Platinum SYBR Green qRT-PCR Kit (Thermo Fisher Scientific, https://www.thermofisher.com). Each reaction was run in triplicate. Primers for the IGT family members were designed using the respective QUIAGEN CLC Genomics Workbench tool (QUIAGEN, https://digitalinsights.qiagen.com/). Actin primers were used as an internal control to normalize expression [56]. The reactions were performed using a 7900 DNA sequence detector (Thermo Fisher Scientific, https://www.thermofisher.com). In ten out of the previous fourteen cultivars (Table 1), an expression analysis for selected transcription factors (TFs) was performed in SGIker, UPV/EHU (Bizkaia, Spain) using a 48*48 Fluidigm array. Primer for the selected transcription factors (TFs) were designed using the online tool Primer3Plus [57]

(http://www.bioinformatics.nl/cgi-bin/primer3plus/primer3plus.cgi). Reactions were carried out using the Fluidigm BioMark HD Nanofluidic qPCR System combined with a GE 48*48 Dynamic Arrays (Fluidigm, https://www.fluidigm.com) and detection through EvaGreen fluorochrome (Bio-Rad Laboratories, https://www.bio-rad.com). CTs were obtained with Fluidigm Real-Time PCR Analysis Software version 4.1.3 (Fluidigm, https://www.fluidigm.com).

### Promoter analysis

The promoter sequences of *LAZY1* and *LAZY2* genes, 1500–1800 bp upstream of the start codon, were analyzed in search of regulatory cis-elements. PlantCARE [58] (http://bioinformatics.psb.ugent.be/webtools/plantcare/html/) and New PLACE [59] (https://www.dna.affrc.go.jp/PLACE) were used to identify putative cis-elements and their correspondent binding factors.

### Statistical analysis

Three biological replicates from different branches of the same tree were used. All the statistical analysis was carried out in R (https://cran.r-project.org/). Analysis of significance for expression analysis was performed using Kruskal-Wallis H test and comparison between means was performed with a Nemenyi test using the PMCMR R package [60].

## Results and discussion

### Prunus dulcis IGT family members

Six IGT family members were found in *P. dulcis* using BLASTp to search homologues from *P. persica* sequences. The *P. persica* nomenclature [61] was kept for *P. dulcis*: *LAZY1* (Prudul26A025589), *LAZY2* (Prudul26A030030), *DRO1* (Prudul26A032079), *DRO2* (Prudul26A028716), *IGT-like* (Prudul26A033016) and *TAC1* (Prudul26A020993). The phylogenetic analysis also revealed that LAZY1 and LAZY2 peptide sequences are closely related, as well as DRO1 and DRO2. TAC1 is more similar to the rest of the members than IGT-like even without the CCL domain (Fig 1, S2 File). Although little is known about IGT-like function, the high variability could suggest a less-essential activity, or at least less selective pressure on its amino acid sequence. DRO1 and DRO2 are the most conserved members among cultivars; DRO1 shares the same protein sequence for all the different cultivars and wild species (Fig 1, S2 File). Despite the fact that polymorphisms are observed trough the different cultivars, overall, the protein sequences of the IGT Family members are highly conserved, hinting to an essential role in tree architecture regulation (Fig 1, S2 File).

### IGT family protein sequence

IGT family proteins share five conserved regions in Arabidopsis, with the exception of TAC1, which lacks the CCL domain in the 3' terminal, which comprise region V (Fig 2). While Regions I, II and V are remarkably conserved, regions III and IV differed more between members, which might indicate that their preservation is not as essential to keep their activity [33]. Furthermore, functional analysis in transgenic rescue experiments involving AtLAZY1 have shown that even proteins with mutated residues in these two regions are able to rescue the *Atlazy1* branch angle phenotype [62]. In *P. dulcis*, a similar display of conserved regions can be seen, with Regions I, II and V extremely conserved while more variability is observed in Regions III and IV (Fig 2). The high degree of conservation that these regions keep throughout plant species highlights its importance in plant regulation.

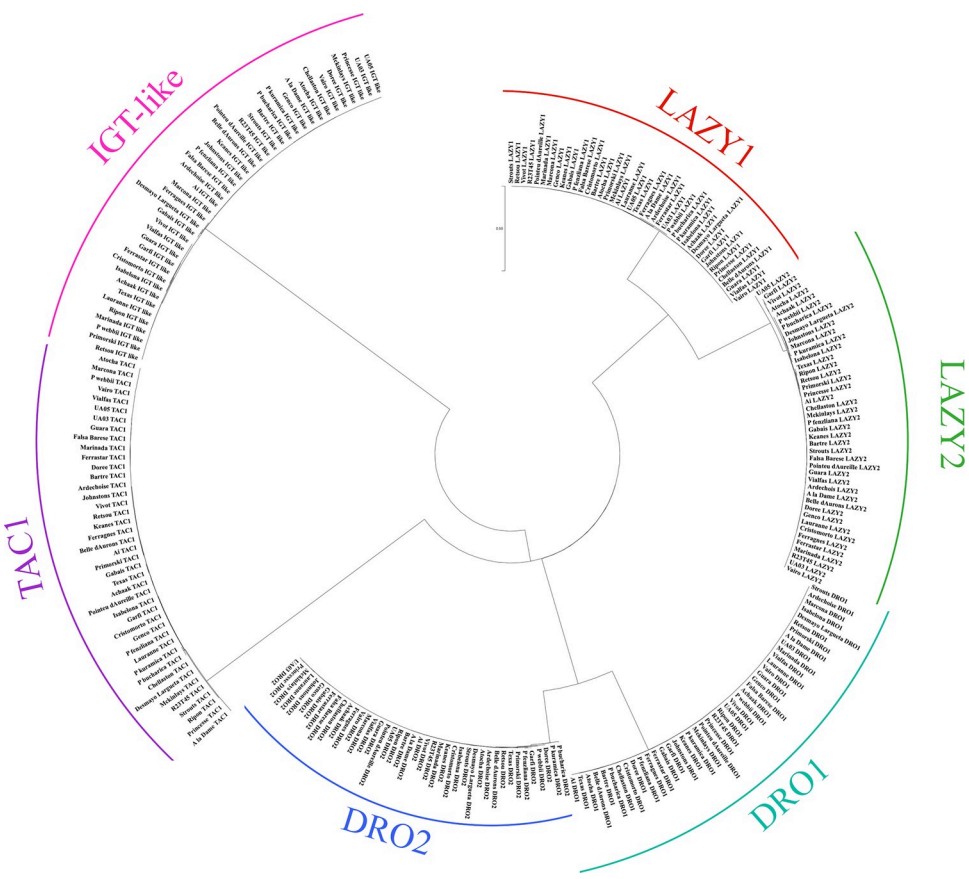

**Fig 1. Phylogenetic tree of the six IGT family in forty-one cultivars and almond wild species.** Cultivars are separated into groups by IGT family protein. Only variants in homozygosis were used for tree building. Names and recorded phenotype of each cultivar and wild species are available in S1 File, while protein sequences can be found in S2 File.

Both LAZY1 and LAZY2 present mutated residues located in conserved regions in several cultivars and wild species. LAZY1 presents a mutation in Region I, I7 is replaced by a methionine (Table 2). Yoshihara and Spalding [62] reported that individuals with the residues 6 to 8 mutated showed significantly reduced ability to rescue the *atlazy1* branch angle defect nor they were able to mobilize the protein correctly to the plasma membrane in Arabidopsis. Therefore, this region seems to be essential for the correct functionality of the signal peptide. However, AtLAZY1 also presents a methionine in this position on the functional protein and the residue can be found mutated in other members of the IGT family, while W6, probably the

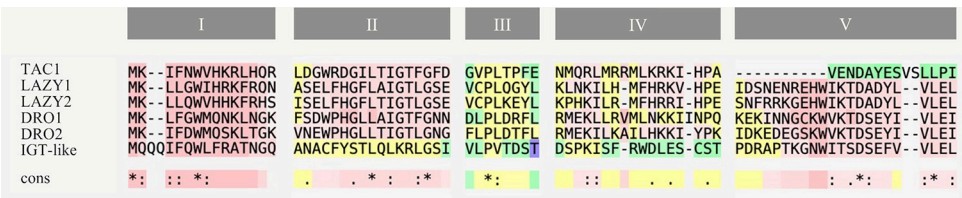

**Fig 2. Amino acid sequence alignment of the five conserved regions between members of the IGT Family in *P. dulcis*.** Sequence alignment analysis was performed using T-COFFEE [63]. Red indicates higher levels of conservation. Sequences from Texas cultivar were used as model (S2 File).

**Table 2. List of mutations of interest whether by their localization or by their predicted outcome.**

| Protein | Mutation | Prediction | Cultivars presenting the variant |
|---------|----------|------------|----------------------------------|
| LAZY1 | I7M | Neutral | 'Bartre' (1), 'Marinada' (2), **'Garfi' (2)**, **'Achaak' (2)**, 'Atocha' (2), **'Princesse' (2)**, *P. kuramica* **(2)**, 'Lauranne' (3), 'Marcona' (3), **'Vialfas' (3)**, 'Vivot' (3), **'Vairo' (3)**, 'Retsou' (3), **'Chellaston' (3)**, **'Isabelona' (3)**, *P. bucharica* **(3)**, 'Guara' (4), 'Primorski' (4), 'Cristomorto' (4), 'Ai' (4), **'Belle d'Aurons' (4)**, 'Genco' (4), 'Pointeu d'Aurielle' (4), **'Desmayo Largueta' (5)** |
| LAZY1 | P18Q | Deleterious, codon change | 'Lauranne' (3), 'Vialfas' (3), 'Vairo' (3), 'Chellaston' (3), 'Guara' (4), 'Ai' (4), 'Belle d'Aurons' (4) |
| LAZY1 | I182_G184del | Deleterious, codon deletion | *P. bucharica* **(3)** |
| LAZY2 | A134E | Deleterious, codon change | 'Bartre' (1), 'Ardechoise' (2), **'Garfi' (2)**, **'Atocha' (2)**, 'Princesse' (2), 'Lauranne' (3), 'Vialfas' (3), **'Vivot' (3)**, 'Retsou' (3), 'Guara' (4), 'Primorski' (4),' Belle d'Aurons' (4), 'Genco' (4) |
| LAZY2 | R293G | Deleterious, codon change | 'Bartre' (1), 'Achaak' (2), **'Marcona' (3)**, 'Chellaston' (3), 'Isabelona' (3), 'Ai' (4), *P. webbii* (4), **'Desmayo Largueta' (5)** |
| TAC1 | D105_D108del | Neutral | *P. bucharica* (3) |
| TAC1 | D108_E109insD | Deleterious, codon insertion | 'Bartre' (1), 'Marinada' (2), 'Ardechoise' (2), 'Achaak' (2), 'Ferragnes' (2), 'Princesse' (2), *P. kuramica* (2), 'Marcona' (3), 'Vialfas' (3), 'Vivot' (3), 'Vairo' (3), 'Retsou' (3), 'Chellaston' (3), *P. bucharica* (3), 'Guara' (4), 'Primorski' (4), 'Ai' (4), 'Belle d'Aurons' (4), 'Pointeu d'Aureille' (4), *P. webbii* (4), 'Desmayo Largueta' (5) |

Only cultivars presenting the mutation are reported. Overall tree habit description is displayed after each cultivar: (1) = Upright, (2) = Somewhat upright, (3) = Semi-open, (4) = Open, (5) = Weeping. Cultivars in bold present the mutation in both alleles. Complete protein sequences for LAZY1, LAZY2 and TAC1 can be found in S2 File. All found variants are listed in S3 File.

indispensable residue, is conserved throughout the members of the family, both in Arabidopsis and almond. This fact would explain why the I7M mutation in homozygosis is not correlated with the observed overall tree habit amongst cultivars (Table 2). Several cultivars present a mutation in the Region IV of LAZY2, replacing R293 for a glycine, although no relation with their phenotype was established. As described by Nakamura *et al.* [33], conservation of Region IV is not required to maintain protein functionality.

A repetitive region of aspartic residues in TAC1 has been previously described as influential in the protein functionality. Differences in their length may lead to effects in the tree architecture; those who have long runs of aspartic acid residues presented upright phenotypes. Additional residues could affect the functionality or stability of the protein [40]. Two different mutations can be observed in our almond cultivars. While a number of cultivars carry the insertion of an additional Asp residue, a deletion of four Asp amino acids can be observed in the wild species *Prunus bucharica*. Nonetheless, in both cases the mutations are presented only in heterozygosis, thus this might explain why no phenotypic variations are observed (Table 2). No mutations in conserved regions were observed for DRO1 and DRO2. This lack of alterations in their sequence can be explained because *DRO1* and *DRO2*, unlike *LAZY1* and *LAZY2*, are described to act mainly in roots [30]. Yet, cultivars are predominantly selected by other aerial traits, such as fruit quality or yield, not existing any artificial selection of favored polymorphisms for tree architecture. The high variability observed in the IGT-like protein sequence combined with unknown function hinders the possibility to discern if any mutated amino acid could affect its activity. After an in-silico analysis using PROVEAN [64] and SNAP platforms (Rostlab, https://www.rostlab.org/) other SNPS and indels were highlighted as possible effectors of phenotypic variance. These were marked as deleterious by these online tools, though their effects were limited to a single codon change, deletion or insertion (Table 2). Moreover, no relation between these mutations and the described phenotypes was observed.

It was not possible to establish a relation between the sequence variants and the diversity in overall tree habit, even though mutations in conserved regions were detected in LAZY1 and

LAZY2 (Table 2), which correlate with previous studies indicating a relatively highly conserved structure for these proteins [33, 36]. In other species, mutations altering the phenotype produced a truncated protein or altered entire exons affecting protein functionality [61]. In our case, there are mutations modifying the protein sequence, however, none of them seem to lead to significant phenotypic impacts. In other herbaceous species these mutations lead to severe effects in cell wall structure that might be even more severe in tree, such as making the individuals that present these variants to be non-viable [61]. However, the difference in tree architecture might be related to quantitative variation of gene expression. To assess this, the expression of IGT family members was analyzed for a group of fourteen selected cultivars, in order to discover if the phenotypic differences could be due to its expression profile.

## Expression profiling of IGT family members in selected almond cultivars

The expression levels of the six IGT family members were analyzed in shoot tips of fourteen almond cultivars on late August (Table 1). Expression analysis could provide an estimation of the protein activity. Previous studies in *P. persica* have shown than *LAZY1* and *TAC1* expression patterns are similar and both genes are expected to be coordinately regulated [31, 35, 41]. Since *TAC1* is believed to act antagonistically to *LAZY* activity, it could be that high levels of *LAZY1* or *LAZY2* expression were influenced by high levels of *TAC1* expression, or vice versa. Furthermore, in poplar (*Populus trichocarpa*), *TAC1* overexpression has been linked to broad-crown trees, while *LAZY1* expression remained constant through both narrow-crown and broad-crown trees [65]. Therefore, we used the *LAZY1*/*TAC1* and *LAZY2*/*TAC1* expression ratio as a descriptor of *LAZY1* and *LAZY2* molecular activity (Fig 3).

*LAZY1*/*TAC1* and *LAZY2*/*TAC1* did show differences in their ratio profile between cultivars. *LAZY1*/*TAC1* was found to have a higher ratio in 'Garnem' shoot tips, while upright cultivars 'Bartre' and 'Ferragnes' had the lowest levels of *LAZY1*/*TAC1* ratio. Other cultivars like 'Garfi', 'Vialfas' and 'Vairo' also presented relatively elevated *LAZY1*/*TAC1* ratios (Fig 3A). Highest levels of *LAZY2*/*TAC1* expression ratio were found in 'Garfi' and 'Vialfas', although the ratio in 'Garfi' was almost 2-fold higher. Unlike 'Garfi', *LAZY2* was not overexpressed in 'Vialfas' compared to the rest of cultivars, yet its lower levels of *TAC1* could indicate an imbalance in the *LAZY2*/*TAC1* ratio and, therefore, a higher *LAZY2* activity. 'Marcona' and 'Vairo' presented the lowest levels of the *LAZY2*/*TAC1* ratio (Fig 3B). It was not possible to find any

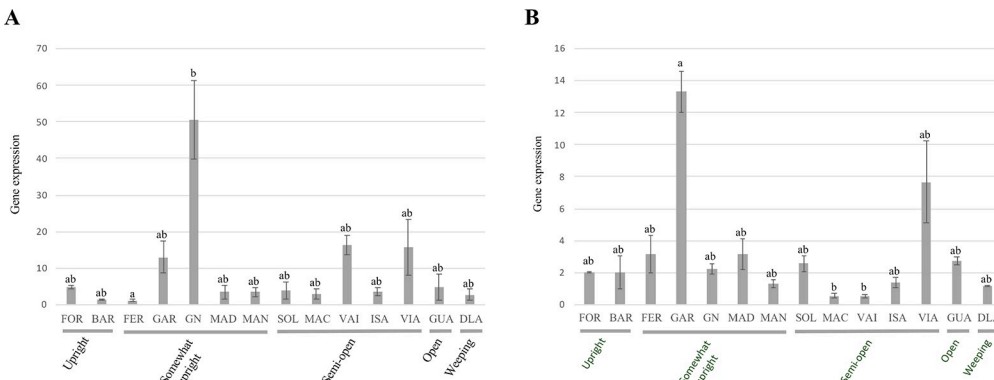

**Fig 3. Expression analysis of IGT family genes in fourteen cultivars of interest.** A, Ratio of relative gene expression between *LAZY1* and *TAC1*. B, Ratio of relative gene expression between *LAZY2* and *TAC1*. Cultivars abbreviatures are as follows: 'Forastero' (FOR), 'Bartre' (BAR), 'Ferragnes' (FER), 'Garfi' (GAR), 'Garnem' (GN), 'Diamar' (DIA), 'Marinada' (MAN), 'Soleta' (SOL), 'Marcona' (MAC), 'Vairo' (VAI), 'Isabelona' (ISA), 'Vialfas' (VIA), 'Guara' (GUA), 'Desmayo Largueta' (DLA). Letters above each bar indicate significance group, derived from Nemenyi's Test.

transcripts of *DRO2* and *LAZY-like*, while *DRO1* expression was only detected in a reduced number of cultivars. This result is not unexpected, since *DRO* genes have been described acting mainly in root tissues [30].

'Garnem' is the only selection that is not a scion cultivar, but rather a hybrid peach x almond rootstock [66]. It has been described that the effect of IGT family members can vary within Prunus species, e.g., *TAC1* silencing in plum (*Prunus domestica*) mimicking the pillar peach genotype leads to more acute effects on tree architecture [40]. The peach genetic background in 'Garnem' could explain why the *LAZY1/TAC1* ratio levels are significantly higher compared to the rest of the analyzed genotypes. 'Garfi', the mother genotype of 'Garnem' shows a similar tree habit phenotype but different expression pattern. In 'Garfi', *LAZY1/TAC1* ratio is moderate and *LAZY2/TAC1* is elevated when compared with the rest of cultivars (Fig 3). However, 'Garfi' expression levels, while being higher than most cultivars, are quite similar for both members of the IGT family, presenting similar absolute values both ratios.

Although significant differences in gene expression were found on branches that presented vegetative growth, it was not possible to establish a correlation between expression levels and overall tree habit in these cultivars. Both 'Garfi' and 'Garnem' present an upright architecture, which would be tied to an expected predominance of *LAZY* expression. However, trees with more erect habits as 'Forastero' and 'Bartre' showed low or basal levels of *LAZY/TAC1* ratios. Expression levels of both *LAZY1* and *TAC1* in *P. persica* have been described to be related to seasonal changes, being higher in April [41]. However, they are expected to be expressed in any growing and active tissue [31]. In Mediterranean areas, almond displays vegetative growth through late spring to end of summer [43]; hence presenting an active growth in its shoot tips during this period. Even though high levels of *LAZY1* and *LAZY2* are presented exclusively in upright cultivars, it does not appear to be the only factor in shaping the almond tree habit, since cultivars with lower ratios present a more upright phenotype. It is possible that the ratio values changes are too low to observe an effect in the phenotype. In poplar, differences that led to a contrasting phenotype were at least an order of magnitude higher to those observed here [65]. Though high similarity has been reported between peach and almond genomes [42, 67], we did not observe in our set of cultivars the effect on the phenotype that has been described in peach [31, 40, 41]. The lack of correlation observed in the studied phase between gene expression and phenotype accompanied by the same case observed with their protein sequence hints to the IGT family may have suffered little to no selection at all in commercial almond orchards (Table 2, Fig 2). Not being unexpected since, until recently, almond breeding has been focused on improving traits related to either flowering or the fruit [68]. Thus, other regulatory pathways must be involved in the establishment of the overall tree habit.

## Analysis of variants in *LAZY1* and *LAZY2* promoter regions

Although it is not possible to establish any clear correlation between diversity in tree habit and the expression levels of the IGT family members, the difference in *LAZY1* and *LAZY2* expression between the related 'Garfi' and 'Garnem' gives us a unique opportunity to study in detail the mechanisms involved in regulating their gene expression. Since these two selections present different expression profiles while their sequences are highly similar, divergences in their promoter region and their transcription factors (TFs) binding capabilities could explain the contrast in expression.

Promoter regions of *LAZY1*, *LAZY2* and *TAC1* were analyzed in search of variants within regulatory elements (REs) that might impact their expression and their respective ratios. Two mutations that could explain the differences observed in their expression profile were found in

**Table 3. List of variants that correlate with the differences observed in gene expression affecting Regulatory Elements (REs) and their Transcription Factors (TFs) associated.**

| Gene | Position | RE | TF | Sequence | Alternative | Cultivars presenting the variant |
|------|----------|-----|-----|----------|-------------|----------------------------------|
| *LAZY1* | Pd01:20652273 | ABRE | *ABI3* | GCCATTTGTC | GCCATTCGTC | 'Bartre' (1), **'Ferragnes' (2)**, 'Marinada' (2), 'Soleta' (3), 'Marcona' (3) |
| *LAZY1* | Pd01:20652273 | E-Box | *RAVL1* | GCCATTTGTC | GCCATTCGTC | 'Bartre' (1), **'Ferragnes' (2)**, 'Marinada' (2), 'Soleta' (3), 'Marcona' (3) |
| *LAZY1* | Pd01:20652307 | TGGGCY-motif | *IPA1* | AGCCCA | GGCCCA | 'Bartre' (1), **'Garnem' (2)**, **'Isabelona' (3)**, 'Guara' (4), 'Desmayo Largueta' (5) |
| *LAZY2* | Pd03:23958144 | GTAC-motif | *IPA1* | GATAAGC | GATAAG | 'Forastero' (1), 'Bartre' (1), **'Garfi' (2)**, 'Garnem' (2), 'Diamar' (2), **'Soleta' (3)**, **'Vialfas' (3)** |

Only cultivars presenting the mutation are reported. Overall tree habit description is displayed after each cultivar: (1) = Upright, (2) = Somewhat upright, (3) = Semi-open, (4) = Open, (5) = Weeping. Cultivars in bold present the mutation in both alleles. All mutations in promoter sequences can be found in S4 File.

*LAZY1* and only one in *LAZY2* (Table 3). No significant variants were encountered in the *TAC1* promoter region.

Both *LAZY1* and *LAZY2* promoter regions presented a variant within a RE which is associated to the TF *IPA1* (Table 3), also known as *SPL9* in *A. thaliana* and *SPL14* in *O. sativa*. *IPA1* has been previously related with the regulation of shoot branching, acting predominantly repressing gene expression, though it has been described to also act in a promoting manner in few cases [69, 70]. In Arabidopsis, it has been reported that *IPA1* downregulates genes involved in responses related to auxin signaling [71]. While *LAZY1* promoter region presents the variant in a TGGGCY motif, *LAZY2* has a mutated GTAC motif (Table 3). *IPA1* has been described to interact with both motifs, and more specifically, directly with the second one [71]. Due to the nature of *IPA1* activity, it would be conceivable that it is acting in a repressive fashion. Therefore, if a mutation obstructs its binding to a RE, *LAZY1* and *LAZY2* would predictably be overexpressed. The mutations described might fit with this predicted outcome, especially in the *LAZY1* promoter region, where 'Garnem' presented the mutation, which displayed a remarkable high *LAZY1*/*TAC1* ratio due to an overexpression of *LAZY1* (Fig 3, Table 3). 'Garfi' also presented a mutation in the *LAZY2* promoter, which could be linked to its elevated *LAZY2*/*TAC1* ratio, though similar levels are observed in *LAZY1*/*TAC1* ratio where no mutation was described (Fig 3, Table 3). Nevertheless, other cultivars also present the variant in this RE without showing high ratio values, indicating that the mutation does not affect gene expression by itself, possibly being affected by other factors, i.e., *IPA1* expression level, protein activity or the interaction of other TFs.

Another mutation of interest was found in the *LAZY1* promoter region, affecting an E-box element, which has been described as a binding region of the transcription factor *RAVL1* (Table 3). The mutation exists in several selected varieties and is present in homozygosis in the cultivar 'Ferragnes' (Table 3), whose *LAZY1*/*TAC1* ratio was low (Fig 3). In rice, *RAVL1* has been described directly promoting genes involved in BRs and ET responses, acting in diverse metabolic processes [72, 73]. BRs act promoting branching and shoot growth [74]. The involvement of *RAVL1* in regulating *LAZY1* and therefore, gravity response, would place this gene at the crossover between both responses. Moreover, an ABRE element described as a binding region for the TF *ABI3* could be also altered by the same mutation. Nevertheless, *ABI3* is mainly involved in ABA signaling and predominantly in processes related to seed germination [75].

The mutations described in *LAZY1* and *LAZY2* promoter might explain the differences in their gene expression through cultivars. In particular, a mutation within a RE related to the TF *IPA1* in the *LAZY1* promoter may cause the high *LAZY1*/*TAC1* ratio observed in 'Garnem'.

Other mutations could also affect the expression profile, though more knowledge is needed to characterize their effect.

## Analysis of expression *IPA1* homologues in *P. dulcis*

Due to its possible involvement in the regulation of *LAZY1* and *LAZY2* expression, a BLASTp search for IPA1 homologues in *P. dulcis* was conducted using atIPA1. Three *IPA1* homologues were found: *IPA1-like 1* (Prudul26A025211), *IPA1-like 2* (Prudul26A009750) and *IPA1-like 3* (Prudul26A016898). No non-synonymous mutations were found for any of the homologues. The expression levels of the three genes were analyzed in the shoot tips previously collected at the end of summer, in ten of the previous fourteen cultivars.

The expression profile through the ten cultivars was relatively stable for the three genes. Cultivars 'Vairo', 'Marinada' and 'Diamar' presented the highest expression levels (Fig 4). However, significant differences were only found in *IPA1-like 2*, which is overexpressed in 'Vairo' and repressed in 'Garfi'. In all three homologues, 'Garfi' presented low expression levels compared with the rest of cultivars. A similar profile can be observed in 'Vialfas' (Fig 4). As it is mentioned before, *IPA1* has been previously described acting as a repressor [69–71]. Therefore, the relative high ratio observed in both *LAZ1/TAC1* and *LAZY2/TAC1* in 'Garfi' might be associated with low *IPA1* activity. Although 'Vialfas' high *LAZY2/TAC1* ratio was mostly explained by *TAC1* repression, a similar phenomenon could underlie its profile. Nonetheless, no REs associated to *IPA1* were found in the analysis of the *TAC1* promoter.

'Garnem' showed similar expression levels that other cultivars for all three *IPA1* homologues, while displaying a remarkably high *LAZY1/TAC1* ratio. This overexpression could be caused by the mutation previously described in the *LAZY1* promoter, affecting a regulatory element associated to *IPA1* regulatory activity (Table 3). The mutation could disrupt *IPA1* interaction with the *LAZY1* promoter, and hence preventing *LAZY1* inhibition (Figs 3 and 4). Since no alterations were found in the *LAZY2* promoter, *IPA1* would be able to repress its expression, leading to the lower *LAZY2/TAC1* ratio observed in 'Garnem'.

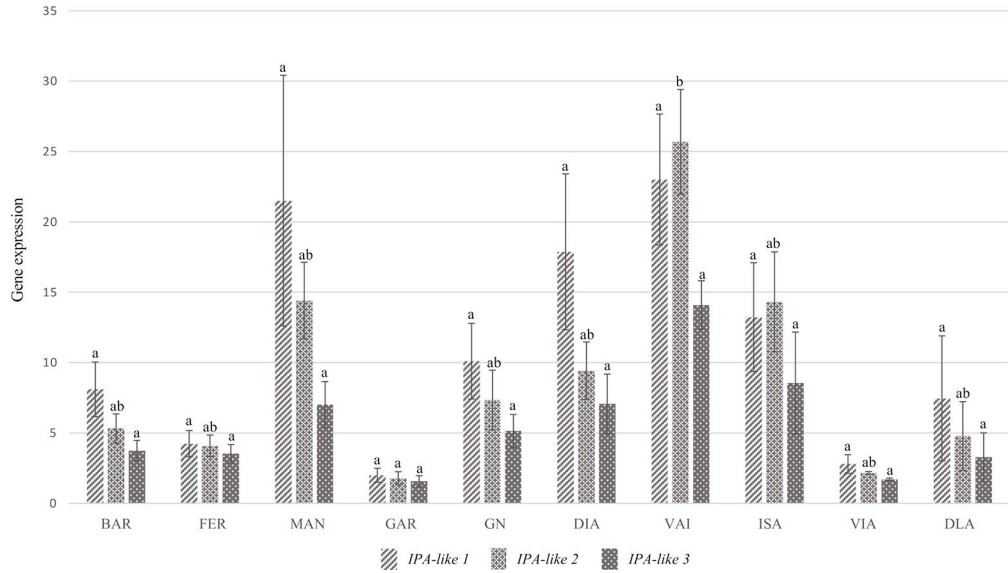

**Fig 4. Expression analysis of *IPA1* homologues in *P. dulcis*.** Cultivars abbreviatures are as follows: 'Bartre' (BAR), 'Ferragnes' (FER), 'Marinada' (MAN), 'Garfi' (GAR), 'Garnem' (GN), 'Diamar' (DIA), 'Vairo' (VAI), 'Isabelona' (ISA), 'Vialfas' (VIA), 'Desmayo Largueta' (DLA). Statistical analysis was performed for each gene separately. Letters above each bar indicate significance group derived from Nemenyi's Test.

*IPA1* homologues seem to act redundantly, presenting a similar expression profile for the three genes. As it can be observed in 'Garfi' and 'Vialfas', low expression levels may be behind high *LAZY1/TAC1* and *LAZY2/TAC1* ratios. Therefore, confirming *IPA1* genes as candidate repressors of *LAZY1* and *LAZY2* activity in *P. dulcis*.

## Regulatory elements and transcription factors in *LAZY1* and *LAZY2* promoter regions

In order to identify TFs that might interact with REs present in *LAZY1* and *LAZY2* promoter regions, these regions were analyzed using New PLACE and PlantCARE online platforms. Twenty-one TFs were selected as preferred candidates, in addition to the previously described *RAVL1* and *ABI3*, which possible RE variability was noted within the varieties (Table 4). A majority of the TFs are involved in light responses and hormonal regulation. Similar functions have been described in the REs of *LAZY1*, *LAZY2* and *TAC1* in *Malus* x *domestica* [76].

Several TFs are involved in auxin responses. While *ARF1* REs are present in both promoter regions, *ARF2* and *IAA24* REs only are found in *LAZY1* promoter; all of them act as mediators in the auxin signaling pathway [77–82]. Other hormone regulatory pathways are represented among the TFs selected. *RAP2.2* and *RAP2.3* belong to the Group VII of ERF (Ethylene Response Factors) and are involved in various stress responses [83–86]. *RAP2.2* REs can be

**Table 4. Localization in the LAZY1 and LAZY2 promoters of identified Transcription Factors (TFs).**

| Transcription factor | P. dulcis ID | Position *LAZY1* | Position *LAZY2* |
|---|---|---|---|
| *ABI3* | Prudul26A014736 | | -1314, -1166, -882, -878, 85 |
| *ARF1* | Prudul26A011950 | -1423 | -1138, -474, 222 |
| *ARF2* | Prudul26A008717 | -1298, -344, -343 | |
| *ATAF1* | Prudul26A030564 | -1299, -345, -344 | |
| *GATA14* | Prudul26A008840 | -33 | -1569, -129 |
| *GBF6* | Prudul26A015068 | -345 | |
| *GTL1* | Prudul26A008868 | -892, -890 | |
| *HB4* | Prudul26A018199 | -1325, -1152 | -1475, -1314, -1102, -882, -878, 85 |
| *HB5* | Prudul26A009108 | | -1246, -1011, -758, 115 |
| *IAA24* | Prudul26A021243 | -678 | |
| *LEAFY* | Prudul26A028984 | 85 | |
| *MYC2* | Prudul26A013616 | -1474, -1296, -1325, -841, -777, -699, -418, -392, -340, -238, -223, -155 | -1413, -908, -672, -304, -284, -164, 404 |
| *OBP4* | Prudul26A018122 | -869, -863 | -1475, -1469, -516 |
| *PCL1* | Prudul26A032278 | | -1139, -744, -743 |
| *phyA* | Prudul26A016497 | | -559 |
| *RAP2.2* | Prudul26A031706 | -1454, -1420, -1374, -1370, -1290, -1203, -1120, -1111, -1046, -1023, -1019, -954, -802, -768, -719, -643, -518, -445, -420, -394, -361, -308, -291, -287, -269, -212, -180, -176, -112, -84, -35, -28, -18, 43, 58, 63, 75, 144, 280, 326' | -1619, -1564, -1267, -1257, -1232, -1113, -1105, -1069, -982, -975, -967, -949, -916, -894, -861, -747, -704, -692, -647, -604, -544, -502, -490, -483, -470, -416, -400, -384, -355, -353, -344, -289, -278, -257, -218, -211, -207, -195, -172, -124, -99, -82, -70, -64, -58, -51, -47, -18, 46, 149, 204, 296, 343, 385, 410 |
| *RAP2.3* | Prudul26A030616 | -1036, 8 | -1090, -236 |
| *RAVL1* | Prudul26A026729 | -779, -157, 87, 85 | -1439, -1277, 402, 402, 402, 403 |
| *SGR5* | Prudul26A008399 | -1426 | |
| *TGA1* | Prudul26A032960 | -1168 | -58 |
| *WUS* | Prudul26A011412 | | 82 |

Position is displayed as relative to the start codon.

found extensively repeated through both promoter regions. *LAZY2* promoter exhibits REs for *HB5*, a positive regulator of ABA and GA responses, and *WUS* a promotor of meristem proliferation in response to ET and auxin [87–89]. The *ATAF1* RE, that falls within the *LAZY1* promoter, is a key regulator of biotic and abiotic stress pathways, promoting ABA biosynthesis and regulating carbon metabolism genes or inducing the expression of genes involved in salt stress and detoxification responses [90–93]. Both promoters have REs for the TF OBP4, which is a negative regulator of cell expansion and root growth in response to ABA [94–96]. *GBF6* with a RE in *LAZY1* promoter, is repressed by sucrose and acts as a mediator between carbohydrates regulation and amino acid metabolism [97]. Sugars have been described as an essential part of branch outgrowth [11]. *TGA4*, with a RE described in both promoters, acts as a regulatory factor that mediate nitrate responses and induce root hair development in Arabidopsis roots [98, 99]. Light response TFs were also included in the selection. Both *LAZY1* and *LAZY2* promoters present a site for *MYC2* and *HB4*, which are involved in R:FR regulation and shade avoidance response [100, 101]. *PCL1 (*RE found in *LAZY2* promoter), is involved in the circadian clock [102, 103]. *GT-1*, found in both promoters, and its family member *GTL1*, only in *LAZY1*, have been described to modulate various metabolic processes in response to light perception [104]. *LAZY2* promoter presents a RE associated to the photoreceptor *phyA*, core regulator of the R:FR ratio light perception [12–15]. REs for *GATA14*, a zing finger TF belonging to the GATA family, are found in both promoters. GATA family of TFs have been described to integrate growth and light perception in several species [105, 106]. Although *LAZY1* and *LAZY2* have been primarily described as regulators of gravity responses, a lack of known TFs related to gravity perception or responses was found. Only *SGR5*, involved in early stages of shoot gravitropism, could be found in the *LAZY1* promoter [107]. LAZY1 promoter present a RE for *LEAFY*, which is a central regulator of inflorescence development [108]. Flower development and tree architecture has been previously linked in studies in *Malus* x *domestica* [109]. Between the TFs identified, there are a prevalence of genes related to several hormones. This points to IGT family genes being affected by numerous regulatory processes, as it could be expected hence their predicted role in a complex trait like tree habit. Gene expression was analyzed for these twenty-one TFs, not observing a connection between their levels and the previously reported *LAZY1*/*TAC1* and *LAZY2*/*TAC1* ratios (S1 Fig). In any case, this TFs collection influence gene expression and act in regulatory pathways differently, therefore, the lack of a wide correlation might be expected.

## Conclusions

IGT family proteins are highly conserved in *P. dulcis*, especially within the five conserved regions and a limited number of variations found across all cultivars. Though no correlation with architectural phenotypes was observed, LAZY1 and LAZY2 did exhibit mutations with an expected impact on their functionality. In addition, despite differences in their expression profile, there was no direct relation between the overall tree habit and their expression. Although IGT family members are known to play a role in tree growth habit in other species, we do not see evidence of their influence in tree habit variability for a considerable number of almond cultivars. This is probably because no loss-of-function mutation has been selected in the set of forty-one studied major commercial almond cultivar that favor this trait, while those correlating with phenotype observed in other species alter significantly the protein structure. Until recently tree habit has not been an influential trait in almond breeding and these types of mutations were probably never selected. Furthermore, several of the mutations found in almond cultivars are present in heterozygosis, hence they could alter the phenotype if appear in homozygosis and be a foundation for possible future breeding efforts. Anyway, there are

many mechanisms leading to different tree habit, and even though *LAZY1* and *LAZY2* are not discriminant in current almond commercial cultivars, other families of genes must be involved in the regulation of almond tree habit. However, important aspects of the regulation of the IGT family in almond have been characterized. TFs *IPA1-like 1*, *IPA1-like 2*, *IPA1-like 3* seems to play a role in the regulation of *LAZY1* and *LAZY2* expression in addition to other TFs involved in hormonal regulation and light perception. In conclusion, almond tree habit depends on numerous factors, which outlines the necessity to better characterize the regulation of this trait and molecular mechanisms behind it both in almond orchards and other fruit trees.

## Supporting information

**S1 File. List of the 41 almond cultivars and wild species.** Overall tree habit phenotype for each cultivar is described categorically according UPOV guidelines.
(XLSX)

**S2 File. Protein sequences of the six IGT family members in the 41 almond cultivars and wild species.** First entry of each cultivar or wild species contains homozygous variants while second entry contains both homozygous and heterozygous variants.
(FASTA)

**S3 File. List of mutations affecting the protein sequence for the 41 almond cultivars and wild species.** HOM: mutation in both alleles; HET: mutation in only one allele.
(XLSX)

**S4 File. List of mutations affecting the promoter sequence of *TAC1*, *LAZY1* and *LAZY2* for the selected 14 almond cultivars.** HOM: mutation in both alleles; HET: mutation in only one allele.
(XLSX)

**S1 Fig. Heatmap of relative gene expression for identified transcription factors.** TFs are separated into three groups, whether they are expected to interact with both promoters or only one of them. Heatmap was constructed in R (https://cran.r-project.org/).
(TIF)

## Acknowledgments

The authors thank for technical and human support provided by SGIker of UPV/EHU.

## Author Contributions

**Conceptualization:** Álvaro Montesinos, Chris Dardick, María José Rubio-Cabetas, Jérôme Grimplet.

**Formal analysis:** Álvaro Montesinos.

**Funding acquisition:** María José Rubio-Cabetas.

**Investigation:** Álvaro Montesinos.

**Supervision:** Chris Dardick, María José Rubio-Cabetas, Jérôme Grimplet.

**Writing – original draft:** Álvaro Montesinos, María José Rubio-Cabetas, Jérôme Grimplet.

**Writing – review & editing:** Chris Dardick.

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
