## [Decision Letter · Decision Letter 0]

15 Jun 2021

PONE-D-21-15102

Polymorphisms and gene expression in the almond IGT family are not correlated to variability in growth habit in major commercial almond cultivars

PLOS ONE

Dear Dr. Grimplet,

Thank you for submitting your manuscript to PLOS ONE. After careful consideration, we feel that it has merit but does not fully meet PLOS ONE’s publication criteria as it currently stands. Therefore, we invite you to submit a revised version of the manuscript that addresses the points raised during the review process.

The ms PONE-D-21-15102 «Polymorphisms and gene expression in the almond IGT family are not correlated to variability in growth habit in major commercial almond cultivars»

aims at scrutinizing almond tree architecture  through the genetic and genomic regulation on branch angle of IGT gene family.

The topic has commercial interest mainly for selecting the cultivars adapted to the recent intensive conduction of almond orchards.

Negative results, as described in this paper, since  genetics and genomics of “IGT family are not correlated to variability in growth habit in major commercial almond cultivars” are not very common to be presented in scientific papers but they are not less challenge.

However this claim might be taken cautiously since the number of cultivars is limited, sampling used one tree, at only one time point.

The paper was revised by specialists who added comments, suggestions and important questions.

The context of IGT family choice could be more focused;

The M&M Section might give additional information;

 Discussion section can also focus better on the obtained results, and on previous ones related to almond genetic variability.

Besides scientific revision as suggeste above and by the reviewers,  the ms needs overall careful quality edition.

English language needs revision for typos

We look forward to receiving your revised manuscript.

Kind regards,

Sara Amancio

Academic Editor

PLOS ONE

Journal Requirements:

Reviewers' comments:

Reviewer's Responses to Questions

**Comments to the Author**

1. Is the manuscript technically sound, and do the data support the conclusions?

Reviewer #1: No

Reviewer #2: Partly

Reviewer #3: Partly

2. Has the statistical analysis been performed appropriately and rigorously? 

Reviewer #1: N/A

Reviewer #2: Yes

Reviewer #3: Yes

3. Have the authors made all data underlying the findings in their manuscript fully available?

Reviewer #1: No

Reviewer #2: No

Reviewer #3: Yes

4. Is the manuscript presented in an intelligible fashion and written in standard English?

Reviewer #1: Yes

Reviewer #2: Yes

Reviewer #3: Yes

5. Review Comments to the Author

Reviewer #1: The present paper by Montesinos et al., explores the variability in the IGT gene family (sequence and expression level) found in almond cultivars, and attempts to perform an association with tree growth habit (architecture) phenotypes. With the analysis performed authors claim that IGT family does not play a critical role in the control of tree habit, in the commercialized cultivars. This is stated in the abstract and already fells very ambitious and faulty, because it suggests that if these genes were silenced, no difference would be found, which is very difficult to conclude based on the data provided.

I feel that authors followed a very simplistic approach to try to prove their claim, but the paper misses detail and would benefit from the extended analysis of more cultivars (phenotype mostly). In addition the manuscript has several issues which don’t benefit the reviewing process, namely the quality of the figures, the detail in the legends, and claims/conclusions that are not easily taken from the results.

Some of the claims for a no-association between IGT family genes and tree growth habit come from gene expression analysis, of IGT members and related TFs. However, this analysis was conducted with a reduced number of cultivars, and the biological replicates were collected from the same tree. Given the dynamic nature of transcription, the approach followed makes very hard to compare results between genes and cultivars. Also, direct comparison of transcription levels of Transcription factors and candidate targets is not linear since is fails to represent protein activity.

The quality of figure 1 is very bad. Legend is very also very poor in terms of detail. It’s impossible to confirm the assumptions made in the results by the authors. I don’t know how many species were included in the tree, which sequences were used (full protein sequences? conserved domains?,…). Also there are no branch support values. Phylogenetic tree was build using Neighbor-joining, which is currently no well accepted for evolutionary studies as it does not take into account evolutionary models.

It’s not clear how can authors take conclusions such as “Although little is known about LAZY-like function, the high variability could suggest a less-essential activity, or at least less selective pressure on its amino acid sequence.” or “DRO1 and DRO2 are the most conserved members among cultivars”.

The tree has little or no information to make this assumptions. For example, DRO1 has divergent clades since the root to the last branch. This is not expected in case of a conserved family. I doubt the methods used to build this tree.

Figure 2. Assuming that authors analyzed multiple cultivars, they should state which cultivar was used to retrieve the sequences represented in this figure.

Authors, describe and discuss the variability found in protein sequences, always in the context of the Tree phenotype (growth habit) however, these characterization is not described in the results/discussion. In addition this was only performed for 14 varieties, which seems quite reduced in the scope of an association study. This adds to the fact that, as authors mentioned, some genes may appear in heterozygous form in a cultivar, which increases the complexity of the analysis. I think that the paper doesn’t approach the initial objectives with the right (enough) datasets.

Tables are cut. The full content is not accessible

Figure 3 lacks quality. Authors state that they used Actin as internal control for qPCR analysis, and then in Fig.3 they show the ratio between LAZY/TAC. However, I’m puzzled by the values on the y axis in Figure 2. Are LAZY genes always more expressed than TAC? Also, authors used as biological replicates, samples from the same tree. Again this is quite reductive, since it may not represent the variability among trees from the same cultivar (eventually grown in different locations, or collected from different places from the crown). Were the samples between different cultivars collected in the same time of day, in the same day? Can the variability found between cultivas reflect environmental factors? Would the authors expect different results if new samples were collected? This also applies to the analysis of IPA1 genes.

Other remarks:

Description of statistical analysis in the Methods section lacks detail. “Analysis of significance…” Significance of what?

Some spelling correction are needed in:

Line 122: were phenotyped

Reviewer #2: The authors present a study on the polymorphisms in the IGT gene family across 41 almond cultivars, and their expression analysis in a subset of 14 cultivars exhibiting diverse tree habit phenotypes. They found no correlation between the expression profile of the genes in shoot tips during the growing season (LAZY1, LAZY2 and TAC1), but further searched for transcriptional regulators of those genes, including IPA1 homologues and other transcription factors based on the analysis of regulatory elements in the promoter regions. They concluded that IGT family genes do not play a critical role in the control of tree habit in currently commercialized almond cultivars, in contrast to what is observed in other species. This is a relevant subject given the potential interest in including tree habit traits in almond breeding programs.

I have a number of major concerns on the presented results as follows. Given that after the comparative genomic analysis, only 7 mutations of interest have been identified, and that a lot of analyses have been performed based on this information, I wonder why the authors didn´t try to validate the polymorphisms by other methods in a smaller set of cultivars with defined phenotypes and chose instead to analyze the expression profiles of TFs putatively regulating some of the IGT genes. I could not find in reference “42” given by the authors (line 121) info on the genome of the cultivars used here, only 10 almond cultivars have been resequenced in that work and not the 41 used here. Furthermore, I don´t see the relevance of including, for instance, information on cultivars that present unknown (n) phenotype in Table 1. In the light of this, the conclusions taken about the relevance of the IGT genes for almond tree habit based on the findings that “it was not possible to establish a relation between the sequence variants and the overall tree habit” (stated by the authors in lines 256-257) and the fact that some of the polymorphisms are heterozygous, should be revised.

In the same work (reference 42) genetic variability analysis has been performed, can the authors refer to the findings regarding the IGT genes if available? Or any other works in which genetic variability was studied (e.g. Goonetilleke et al 2018, doi:10.1534/g3.117.300376?).

Overall, while the idea behind the study is scientific sound, there is information lacking in Materials and Methods that the authors should provide to improve the quality of the manuscript. The Results and Discussion section is too long and sometimes speculative, based on the presented results, but on the other hand, it does not seem to explore previous data on almond genetic variability. The English is clear but needs a thorough revision, many errors can be found throughout the manuscript.

Other comments

Materials and Methods

Line 121 - Some additional information of the cultivars and wild species used in the comparative analysis of the IGT family protein sequences should be included. In the Results, only the cultivars with variants are shown, and I guess also in Fig. 1, but the definition of the Figure is not good enough to read. An alternative could be to list the names in the legend of Fig.1 if the names are not clearly visible. As mentioned above, also information on the source of the genome data used here should be given.

Line 152 – Again, some more information is needed about the samples used for RNA extraction and qPCR. Were the trees the same age? And growing in similar conditions (e.g. the same field)? A lot of variation can be introduced in the expression quantification by qPCR depending on the samples used. Given that a lot of the discussion is based on the expression analyses, it is absolutely essential that the authors provide further details about the material used. Additionally, actin was used as reference, please give support to the use of this gene as reference.

Results and Discussion

Tables 2, 3, 4 are truncated

In Table 2, the legend refers to mutations of interest only, if any other mutations were found these should be provided in supplemental information.

Line 250 – “lack of function” or “unknown function”?

Line 266 – what were the criteria to select the subset of 14 cultivars for IGT expression analysis?

The discussion is sometimes too speculative for the results shown. Part of the discussion can be considerably shortened. For instance, in lines 359-366 discussion about the TB1 gene is unnecessary because as the authors say, no homologues to TB1 have been found in any dicot.

Line 390 – For the IPA1 genes, only 10 of the 14 cultivars were used. What were again the selection criteria?

Line 391 – the sentence does not seem consistent with the subsequent explanation, please revise.

Figure 4 – the statistical significance does not seem to make sense based on the use of letters a, b and ab, please check

Table 4 – I cannot see any info on the position of phyA and WUS in the table.

Lines 459 and below lines – this section does not add any relevant information, I suggest this may be placed in supplemental information

Typographical or grammatical errors: line 30-share, line 49-increased, line 78-reorganizations, line 114-could be in, line 115-its, line 122-phenotype, line 126-this, line 141-5’ region?, line 218-through, line 294-reduce, line 305-high, line 306-absiolut values both ratios, line 321-Which is not (rephrase), line 334-explained, line 344-have, line 349-specifically directly, line 370-have, line 439-has, line 455-Between, line 507-characterized

Reviewer #3: Academic Editor

PLOS ONE

Dear Prof Sara Amancio

I am writing my comments regarding the following manuscript which has been submitted to PLOS ONE: Manuscript Number: PONE-D-21-15102.

The paper describes the ‘Polymorphisms and gene expression in the almond IGT family are not correlated to variability in growth habit in major commercial almond cultivars’

The authors used IGT family genes to correlate it with tree architecture in almond.

General comments: The manuscript is a primary work to obtain null or negative result that IGT family genes do not play a critical role in the control of tree habit in currently commercialized almond cultivars. However, the paper suffers a major experimental sampling as the authors have used one sampling time at the end of summer to obtain the expression pattern and growth habit. As you know the expression of genes changed spatially and timely. I believe that to obtain such correlation need more sampling from different tissues and during the developmental stages.

However, the paper could contribute good primary information to our knowledge in tree architecture in almond, and I suggest that the ms prepared in the form of short communication or letter or short report.

6. PLOS authors have the option to publish the peer review history of their article (what does this mean?). If published, this will include your full peer review and any attached files.

Reviewer #1: No

Reviewer #2: No

Reviewer #3: No

---

## [Author Response · Author response to Decision Letter 0]

14 Jul 2021

Dear Pr. Amâncio and reviewers,

We have submitted the revised version of the Manuscript “Polymorphisms and gene expression in the almond IGT family are not correlated to variability in growth habit in major commercial almond cultivars”. We have address the comments raised by the reviewers in the following letter and in the manuscript. 

Reviewer #1: 

“The present paper by Montesinos et al., explores the variability in the IGT gene family (sequence and expression level) found in almond cultivars, and attempts to perform an association with tree growth habit (architecture) phenotypes. With the analysis performed authors claim that IGT family does not play a critical role in the control of tree habit, in the commercialized cultivars. This is stated in the abstract and already fells very ambitious and faulty, because it suggests that these genes were silenced, no difference would be found, which is very difficult to conclude based on the data provided.”

Following our results, IGT family does not seem to play a role in the diversity in tree habit observed in almond cultivars, and therefore there are not a promising marker for future breeding effort for this trait and we consider this information of interest since it have been the case in other species. However, this does not mean that these proteins are not part of control tree habit or in a wider extend, in the tree growth and raising, but they are not the limiting factor of this particular trait. Moreover, high protein conservation through almond cultivars points to whether these genes were silenced, the effect could be indeed dramatic or even lethal. An effort to clarify this conclusion has been made in the abstract and further in the text (lines 39, 45, 474), for example, we instead of saying that they play a role in the control we now say that they are not correlated to the diversity of phenotypes. In any case, the goal of this study was to analyze the nuance changes that may explain the existing diversity of tree habit in almond cultivars, searching for a similar effect to the one observed in peach trees with the br mutation.

“Some of the claims for a no-association between IGT family genes and tree growth habit come from gene expression analysis, of IGT members and related TFs. However, this analysis was conducted with a reduced number of cultivars, and the biological replicates were collected from the same tree. Given the dynamic nature of transcription, the approach followed makes very hard to compare results between genes and cultivars.” 

For practical reasons, it is not possible to maintain multiple adult individuals of a same cultivar in experimental orchards. Typically, only a couple of individuals are kept. Nevertheless, we have added some information in Materials and Methods (lines 154-156) about sample collection that indeed was missing in the manuscript. All samples were collected the same day from the same orchard, whose localization is now available in its M&M section, which reduce the variability caused by the environment. We also used a “reduced” number of cultivar because of the phenotyping study that allowed us to filter out cultivar with an intermediate phenotype, which should be taken into account compared to a study that would have been performed on a random selection of cultivars and worked only on the extreme phenotypes. Our experimental design is within the norm of what is normally perform when comparing the expression of a candidate gene in two genotypes with phenotypic difference with the usual 3 biological replicate, instead we incremented the number of genotypes for each phenotype. Again, we are not claiming that for some of the genotypes IGT control the tree habit but we surely did observe that this make it a rather bad marker for MAS.

“Also, direct comparison of transcription levels of Transcription factors and candidate targets is not linear since is fails to represent protein activity.”

It is correct that TF expression levels has its limitations, and it is not a direct connection with protein abundance and TF regulatory activity (line 278). However, it gives us a good approximation to the regulatory interactions between different TFs and pathways, and allows us to identify possible candidates in the regulation of the IGT family. We added a sentence in the manuscript to remind the reader that expression studies are elements for estimation of protein activity.

“The quality of figure 1 is very bad. Legend is very also very poor in terms of detail. It’s impossible to confirm the assumptions made in the results by the authors. I don’t know how many species were included in the tree, which sequences were used (full protein sequences? conserved domains?,…). Also there are no branch support values. Phylogenetic tree was build using Neighbor-joining, which is currently no well accepted for evolutionary studies as it does not take into account evolutionary models. It’s not clear how can authors take conclusions such as ‘Although little is known about LAZY-like function, the high variability could suggest a less-essential activity, or at least less selective pressure on its amino acid sequence.’ or ‘DRO1 and DRO2 are the most conserved members among cultivars.’ The tree has little or no information to make this assumptions. For example, DRO1 has divergent clades since the root to the last branch. This is not expected in case of a conserved family. I doubt the methods used to build this tree.”

We acknowledge that the quality of the figures in the pdf is low, but it is independent from us and the figures can be visualized at the correct resolution by clicking on the links on the upper right side of the figures. Also the pdf generated for biorXiv has better resolution initial submission https://doi.org/10.1101/2021.05.11.443553

A new Figure 1 has been made, using Maximum Likelihood for tree building, although there are only small differences in distances between cultivars with the previous one. The tree is also displayed differently now, to facilitate its comprehension. Previous figure was not in scale, only grouping cultivars with similar or equal sequences but not representing distances between them. That was the case of DRO1 for example, which while did not present any differences in protein sequence, the image had divergent branches. Now, distances are at scale, which highlight the high degree of conservation of almost all members of the IGT family in almond.

The names of the 41 cultivars should be more visible in the new figure, but we have also added a supplementary file (S1 File) with the 41 cultivars and wild species used for tree building and its phenotype (in the case it could be annotated). Moreover, another supplementary file (S2 File) has been added containing the protein sequences for the six members of the IGT family in each of the 41 cultivars and wild species.

“Figure 2. Assuming that authors analyzed multiple cultivars, they should state which cultivar was used to retrieve the sequences represented in this figure.”

Figure 2 legend now clarifies that Texas cultivar was used for this figure. Also, the whole protein sequence for Texas can be now found in S2 File.

“Authors, describe and discuss the variability found in protein sequences, always in the context of the Tree phenotype (growth habit) however, these characterization is not described in the results/discussion. In addition this was only performed for 14 varieties, which seems quite reduced in the scope of an association study. This adds to the fact that, as authors mentioned, some genes may appear in heterozygous form in a cultivar, which increases the complexity of the analysis. I think that the paper doesn’t approach the initial objectives with the right (enough) datasets.”

Assuming that presence of only Table 1 could lead to confusions, we have added a supplementary file (S1 File) with all the individuals used in the tree building and protein sequence association study. Out of the 41 cultivars used, 27 were described for tree habit and utilized in the association study. Their phenotype can be now found in S1 File. Moreover, all variants in their coding region are in S3 File, apart from those of interest described in Table 2. In addition, we have added the p-values obtain after a GLM analysis using Tassel in S3 File. It can be observed that they are so far from reaching an admissible level of significance, it is extremely unlikely that increasing the dataset would have any impact and that one of these mutations could be considered as a good molecular marker. 

“Figure 3 lacks quality. Authors state that they used Actin as internal control for qPCR analysis, and then in Fig.3 they show the ratio between LAZY/TAC. However, I’m puzzled by the values on the y axis in Figure 2. Are LAZY genes always more expressed than TAC?”

Figure 3 have been substituted by other with higher quality. Values of each gene are relative to the expression of one of each sample, therefore Figure 3 represents relative differences between cultivars. Since LAZY and TAC act coordinately but in an opposite manner, LAZY/TAC ratio was introduced to represent both LAZY and TAC gene expression in only one figure, dividing LAZY1 and LAZY2 expression values by TAC1 values. Normally, TAC1 relative values were lower than those of LAZY1 and LAZY2, but as it can be seen by the presence of values below 1 in Marcona (MAC) and Vairo (VAI) in LAZY2, not always. Anyhow, although TAC1 use to be less expressed both in almond and peach, no conclusions were extracted in this case by the fact that this to cultivars presented values for LAZY2 below TAC1 because these are relative levels for each gene.

“Also, authors used as biological replicates, samples from the same tree. Again this is quite reductive, since it may not represent the variability among trees from the same cultivar (eventually grown in different locations, or collected from different places from the crown). Were the samples between different cultivars collected in the same time of day, in the same day? Can the variability found between cultivas reflect environmental factors? Would the authors expect different results if new samples were collected? This also applies to the analysis of IPA1 genes.”

As said before, relevant information about the sample collection has been added to Materials and Methods (lines 154-156).

Description of statistical analysis in the Methods section lacks detail. “Analysis of significance…” Significance of what?

This part (line 187) now reads: “Analysis of significance for expression analysis was performed using Kruskal-Wallis H test and comparison between means was performed with a Nemenyi test using the PMCMR R package.”

 

Reviewer #2:

“I have a number of major concerns on the presented results as follows. Given that after the comparative genomic analysis, only 7 mutations of interest have been identified, and that a lot of analyses have been performed based on this information, I wonder why the authors didn´t try to validate the polymorphisms by other methods in a smaller set of cultivars with defined phenotypes and chose instead to analyze the expression profiles of TFs putatively regulating some of the IGT genes.”

We saw that even we detected mutations in the different cultivars; they were not likely to provoke significant changes in the activity since none correlates with phenotype. Therefore, we did not attempt to try to confirm that they did not associate in another population. We considered that analyzing the expression of possible regulatory factors was a lead that was worth exploring in order to detect a limiting factor for tree habit. Since our results already gave us little hope that we would find some associations between IGT polymorphism and tree habit, we choose to use an alternative approach by going upstream in the signal transduction pathway. For that we studied the expression of the transcription factors for which a regulatory motif was detected in the promotor area.

“I could not find in reference “42” given by the authors (line 121) info on the genome of the cultivars used here, only 10 almond cultivars have been resequenced in that work and not the 41 used here. Furthermore, I don´t see the relevance of including, for instance, information on cultivars that present unknown (n) phenotype in Table 1.”

The rest of resequences are yet to be published by the consortium. Protein sequences in the 41 cultivars used for the analysis have been added in a supplementary file (S2 File). To facilitate results interpretation, cultivars with unknown phenotype have been discarded in Table 2.

“In the light of this, the conclusions taken about the relevance of the IGT genes for almond tree habit based on the findings that “it was not possible to establish a relation between the sequence variants and the overall tree habit” (stated by the authors in lines 256-257) and the fact that some of the polymorphisms are heterozygous, should be revised.”

No clear relation to the phenotype could be established neither with homozygous nor heterozygous polymorphisms. 

“In the same work (reference 42) genetic variability analysis has been performed, can the authors refer to the findings regarding the IGT genes if available? Or any other works in which genetic variability was studied (e.g. Goonetilleke et al 2018, doi:10.1534/g3.117.300376?).” 

To our knowledge there is the first study of genetic variability in a tree species for the IGT family. Moreover, no characterization of genetic variability in almond referring to tree architecture has been previously reported. Although there is a high similarity between almond and peach genomes, we did not observed differences in gene expression correlating to phenotype, as it has been described in peach (line 326-328).

“Line 121 - Some additional information of the cultivars and wild species used in the comparative analysis of the IGT family protein sequences should be included. In the Results, only the cultivars with variants are shown, and I guess also in Fig. 1, but the definition of the Figure is not good enough to read. An alternative could be to list the names in the legend of Fig.1 if the names are not clearly visible. As mentioned above, also information on the source of the genome data used here should be given.”

Figure 1 has been modified for an easier interpretation. Previous one did not present branches at scale. Also, now cultivars names should be more readable. Moreover, we have added a supplementary file (S1 File) with all 41 cultivars and wild species names and the phenotypes for those that could be annotated. Protein sequences used for tree building can be found in S2 File.

“Line 152 – Again, some more information is needed about the samples used for RNA extraction and qPCR. Were the trees the same age? And growing in similar conditions (e.g. the same field)? A lot of variation can be introduced in the expression quantification by qPCR depending on the samples used. Given that a lot of the discussion is based on the expression analyses, it is absolutely essential that the authors provide further details about the material used. Additionally, actin was used as reference, please give support to the use of this gene as reference.”

Relevant information about sample collection that was missing have been added to Materials and Methods (lines 154-156). All samples were collected from 1-year-old branches. Not all trees were the same age, though all of them were adult trees. Cultivars were kept in the same orchard. 

“In Table 2, the legend refers to mutations of interest only, if any other mutations were found these should be provided in supplemental information.”

All mutations in coding regions for the 6 IGT family members are now available in S3 File.

“Line 266 – what were the criteria to select the subset of 14 cultivars for IGT expression analysis?”

These 14 cultivars were selected to both represent the five tree habit categories and commercial cultivars of importance in Spain.

“The discussion is sometimes too speculative for the results shown. Part of the discussion can be considerably shortened. For instance, in lines 359-366 discussion about the TB1 gene is unnecessary because as the authors say, no homologues to TB1 have been found in any dicot.”

This part of the manuscript has been deleted.

“Line 390 – For the IPA1 genes, only 10 of the 14 cultivars were used. What were again the selection criteria?”

Due to practical reasons, we discarded 4 cultivars that were repetitive for both phenotype and expression profile. 

“Line 391 – the sentence does not seem consistent with the subsequent explanation, please revise.”

Now it reads: “The expression profile through the ten cultivars was relatively stable for the three genes. Significant differences were only found in IPA1-like 2, which is overexpressed in ‘Vairo’ and repressed in ‘Garfi’ (Fig. 4).”

“Figure 4 – the statistical significance does not seem to make sense based on the use of letters a, b and ab, please check”

Statistical significance analysis was performed for each gene separately. This has been clarified in the Figure 4 legend.

Lines 459 and below lines – this section does not add any relevant information, I suggest this may be placed in supplemental information

This part has been summed up and added to the end of the previous section, while Figure 5 is now S5 File. 

Reviewer #3:

“General comments: The manuscript is a primary work to obtain null or negative result that IGT family genes do not play a critical role in the control of tree habit in currently commercialized almond cultivars. However, the paper suffers a major experimental sampling as the authors have used one sampling time at the end of summer to obtain the expression pattern and growth habit. As you know the expression of genes changed spatially and timely. I believe that to obtain such correlation need more sampling from different tissues and during the developmental stages.”

While gene expression characterization depends on samples collected in a specific time, protein sequence analysis does not, since it is based on genomic data. No correlation between the variability observed in growth habit and the variability observed in protein sequence could be stablished also in this step. Furthermore, previous experiments analyzing the expression of this gene family were also limited to shoot tips, since not only is where they present higher levels of transcription, but also where they carry out their molecular function.

---

## [Decision Letter · Decision Letter 1]

5 Aug 2021

PONE-D-21-15102R1

Polymorphisms and gene expression in the almond IGT family are not correlated to variability in growth habit in major commercial almond cultivars

PLOS ONE

Dear Dr. Grimplet,

Thank you for submitting your manuscript to PLOS ONE. After careful consideration, we feel that it has merit but does not fully meet PLOS ONE’s publication criteria as it currently stands. Therefore, we invite you to submit a revised version of the manuscript that addresses the points raised during the review process.

Dear authors

You are almost there

In fact I agree with reviewer 3 and ask for:

1) justification of end of summer sampling;

2) a more robust discussion of the eventual absence of crosstalk between the expression of IGT family members and tree habit

We look forward to receiving your revised manuscript.

Kind regards,

Sara Amancio

Academic Editor

PLOS ONE

Journal Requirements:

Reviewers' comments:

Reviewer's Responses to Questions

**Comments to the Author**

1. If the authors have adequately addressed your comments raised in a previous round of review and you feel that this manuscript is now acceptable for publication, you may indicate that here to bypass the “Comments to the Author” section, enter your conflict of interest statement in the “Confidential to Editor” section, and submit your "Accept" recommendation.

Reviewer #2: All comments have been addressed

Reviewer #3: All comments have been addressed

2. Is the manuscript technically sound, and do the data support the conclusions?

Reviewer #2: (No Response)

Reviewer #3: Partly

3. Has the statistical analysis been performed appropriately and rigorously? 

Reviewer #2: (No Response)

Reviewer #3: Yes

4. Have the authors made all data underlying the findings in their manuscript fully available?

Reviewer #2: (No Response)

Reviewer #3: Yes

5. Is the manuscript presented in an intelligible fashion and written in standard English?

Reviewer #2: (No Response)

Reviewer #3: Yes

6. Review Comments to the Author

Reviewer #2: (No Response)

Reviewer #3: Dear Prof Sara Amancio

I am writing my comments regarding the revised manuscript which has been submitted to PLOS ONE: Manuscript Number: PONE-D-21-15102R1.

The authors addressed all the comments by reviewers. Anyhow, I am not satisfied with their response to my comment. Still, I believe that using only one sampling time, at the end of summer to obtain the expression pattern and growth habit, is not enough to made such conclusion. However, I knew working on almond tree are not so easy to take more sampling, but I suggest that the authors add an explanation in M&M and relevant result part that why they have used end of summer to obtain the gene expression. Also, in the result part ‘Expression profiling of IGT Family members in selected almond cultivars’ fifth paragraph the sentence “it was not possible to establish a general pattern between expression levels and overall tree habit in these cultivars” should be deleted or revised since to obtain an expression pattern a greater number of samplings required.

However, the paper could contribute good primary information to our knowledge in tree architecture in almond, and I suggest after minor revision accepted for publication in Plos One.

7. PLOS authors have the option to publish the peer review history of their article (what does this mean?). If published, this will include your full peer review and any attached files.

Reviewer #2: No

Reviewer #3: **Yes: **Behrouz Shiran

While revising your submission, please upload your figure files to the Preflight Analysis and Conversion Engine (PACE) digital diagnostic tool, https://pacev2.apexcovantage.com/. PACE helps ensure that figures meet PLOS requirements. To use PACE, you must first register as a user. Registration is free. Then, login and navigate to the UPLOAD tab, where you will find detailed instructions on how to use the tool. If you encounter any issues or have any questions when using PACE, please email PLOS at figures@plos.org. Please note that Supporting Information files do not need this step

---

## [Author Response · Author response to Decision Letter 1]

26 Aug 2021

Dear Pr. Amâncio and reviewers,

We have submitted the revised version of the Manuscript “Polymorphisms and gene expression in the almond IGT family are not correlated to variability in growth habit in major commercial almond cultivars”. In the following letter and the manuscript we are addressing the point that yourself and the reviewer 3 have raised.

1) justification of end of summer sampling;

2) a more robust discussion of the eventual absence of crosstalk between the expression of IGT family members and tree habit

Reviewer#3: Dear Prof Sara Amancio

I am writing my comments regarding the revised manuscript which has been submitted to PLOS ONE: Manuscript Number: PONE-D-21-15102R1.

The authors addressed all the comments by reviewers. Anyhow, I am not satisfied with their response to my comment. Still, I believe that using only one sampling time, at the end of summer to obtain the expression pattern and growth habit, is not enough to made such conclusion. However, I knew working on almond tree are not so easy to take more sampling, but I suggest that the authors add an explanation in M&M and relevant result part that why they have used end of summer to obtain the gene expression. Also, in the result part ‘Expression profiling of IGT Family members in selected almond cultivars’ fifth paragraph the sentence “it was not possible to establish a general pattern between expression levels and overall tree habit in these cultivars” should be deleted or revised since to obtain an expression pattern a greater number of samplings required.

First, we really appreciated from reviewer 3 that he acknowledges the difficulties for sampling in almond. We choose the time point at the end of summer because it was one that was the likely to give results since the shots are in active growth but we acknowledge that we could have looked at other time points. Since we did not observed differences of expression at that point, which was following all the previous results on the gene structure and polymorphism, we technically could not invest more time and energy in a hypothesis where we have so many evidences that we will not confirm it. 

However we took the opportunity that PlosOne offers the possibility to researchers to publish these kinds of “negative” results and our goal is to signal the scientific community that despite their role in other species, in almond cultivars, the IGT family genomics data and these transcriptomic data do no validate the hypothesis that they play a role in the variability observed in tree habit. 

We also reviewed the discussion according to the reviewer comments on the fact that we did not observed a pattern but expression at a single time point.

---

## [Editor Report · Decision Letter 2]

30 Sep 2021

Polymorphisms and gene expression in the almond IGT family are not correlated to variability in growth habit in major commercial almond cultivars

PONE-D-21-15102R2

Dear Dr. Grimplet

We’re pleased to inform you that your manuscript has been judged scientifically suitable for publication and will be formally accepted for publication once it meets all outstanding technical requirements.

Kind regards,

Sara Amancio

Academic Editor

PLOS ONE
---

## [Editor Report · Acceptance letter]

5 Oct 2021

PONE-D-21-15102R2 

Polymorphisms and gene expression in the almond IGT family are not
correlated to variability in growth habit in major commercial almond cultivars 

Dear Dr. Grimplet:

I'm pleased to inform you that your manuscript has been deemed suitable for publication in PLOS ONE. Congratulations! Your manuscript is now with our production department. 

Kind regards, 

on behalf of

Prof Sara Amancio 

Academic Editor

PLOS ONE